# Evaluation of Reservoir-Induced Hydrological Alterations and Ecological Flow Based on Multi-Indicators

**Mingqian Li [1,2,3], Xiujuan Liang [1,2,3], Changlai Xiao [1,2,3,*], Xuezhu Zhang [4], Guiyang Li [5], Hongying Li [5] and Wenhan Jang [6]**

[1] Key Laboratory of Groundwater Resources and Environment, Ministry of Education, Jilin University, Changchun 130021, China; hydrogeolmq@163.com (M.L.); xjliang@jlu.edu.cn (X.L.)

[2] Jilin Provincial Key Laboratory of Water Resources and Environment, Jilin University, Changchun 130021, China

[3] College of New Energy and Environment, Jilin University, Changchun 130021, China

[4] Jilin Provincial Water Resources and Hydropower Consultative Company, Changchun 130000, China; jdzhangxuezhu@sina.com

[5] Liaoning Provincial Department of Water Resources, Shenyang 110000, China; 13940435180@139.com (G.L.); a15764303789@163.com (H.L.)

[6] Institute of Disaster Prevention Science and Technology, Beijing 100000, China; 13804071450@163.com

\* Correspondence: xiaocl@jlu.edu.cn; Tel.: +86-1350-082-7868

**Abstract:** Although they fulfill various needs of human beings, reservoirs also cause hydrological regime variation in the downstream regions, thus affecting ecological diversity. Therefore, studying the reservoir-induced hydrological alterations and ecological effects is of great significance, as it could guide the regulation of the reservoir to protect the river ecology. In this study, taking the Taizi River as an example, the impact of a reservoir on hydrological alteration and ecological diversity was comprehensively evaluated through eco-flow indicators based on the flow duration curve and multiple hydrological indicators. The results reveal that: (1) Ecological indicators can be used to analyze the annual and seasonal changes in the streamflow after the construction of the reservoir. The high-flow values and frequency decrease after the construction of the reservoir, especially in the autumn, while the low-flow component values increase significantly, especially in spring and summer. (2) The main influencing factors of the ecological indicators can be reflected by the relationship with precipitation, as the annual ecosurplus is not significantly affected by the reservoir, while the ecodeficit is greatly affected, and the seasonal ecological indicators (especially in spring and summer) are greatly affected by the reservoir. (3) The indicators of hydrologic alteration (IHA) show significant changes after the construction of the reservoir and are consistent with the changes in the eco-flow indicators; the change in the Shannon index indicates that the ecological diversity reduced after construction of the reservoir. It is controlled by the reservoir, and a new equilibrium state appears. (4) The eco-flow indicators have a good correlation with the 32 IHAs; they can reflect the change information of most IHAs and can avoid statistical redundancy.

**Keywords:** reservoir-induced; ecosurplus; ecodeficit; hydrological alterations; indicator of hydrological alteration; ecological diversity; flow duration curve

## 1. Introduction

The river system is the aorta of the Earth and plays a key role in the global water cycle, local climate change, and ecological balance. In recent years, the degradation of river ecology has occurred due to

climate change and intensified human activities [1–4]. The most important human activities are the construction of water conservancy projects such as reservoirs and dams, which have an impact on downstream ecology by changing the hydrological regime [5,6]. Globally, about 70% of all rivers are dispersed by large reservoirs [7], and, according to the latest data from the International Commission on Large Dams (ICOLD), there are 57,985 large reservoirs in the world, of which more than 40% are distributed in China. The impact of reservoirs depends mainly on the ratio of their capacity to natural streamflow, functions, and regulation rules [8,9]. While realizing flood control, power generation, water supply, and other functions, the reservoir also blocks the natural flow of rivers, causing changes in hydrological processes and dynamics, which will have a direct impact on river biodiversity and ecological functions [5,10]. Reservoir-induced hydrological alterations and their ecological impact have attracted more and more attention from ecologists, hydrologists, and policy-makers.

In the study of the comprehensive impact of climate change and human activities on aquatic ecology (including rivers, lakes, and wetland), various models are usually used for evaluation or verification, such as water balance model [11], dynamic water and salt budget model [12], and numerical models based on finite difference method [13,14]. However, to evaluate the ecological impacts of the reservoir, indicators are generally required to quantify the extent of hydrological alteration. Olden and Poff summarized more than 170 hydrological indicators, but there exists a clear correlation between them and statistical redundancy [15]. Richter established the Indicators of Hydrologic Alteration (IHA) for assessing hydrological alterations [16], which are widely used. The IHAs are divided into five categories: monthly mean flow, magnitude and duration of annual extreme flow, timing of annual extreme water, frequency and duration of high and low pulses, and rate and frequency of flow condition changes. There are 32 indicators, which were later revised to 33 [17]. Based on the IHA, a series of methods such as the range of variability method (RVA) and the Dundee hydrological regime alteration method (DHRAM) were developed to quantitatively assess the degree of hydrological alteration and the ecological risk to rivers of hydrological alterations [18–20].

Compared with the 170 indicators, the 33 IHAs are simplified, but the correlation between the indicators has not been well resolved, which may complicate the assessment of ecological flows [21–23]. Vogel proposed dimensionless eco-flow indicators based on the flow duration curve (FDC) and the ecodeficit and ecosurplus, which can directly reflect the surplus or deficit of river ecological flows [24]. In this work, these two indicators are called eco-flow indicators. The introduction of the ecodeficit and ecosurplus provides a new direction for hydrological research under the influence of reservoirs; however, owing to the limited research on this topic, it is not clear whether the eco-flow indicators can better reflect the degree of change of streamflow time series. Furthermore, the relationship between these eco-flow indicators and traditional hydrological indicators is also unclear.

In addition, the research on hydrological alteration and eco-flow under the influence of reservoirs mainly focuses on large river basins [21,22,25,26]. Large-scale watersheds usually span multiple climatic zones and are characterized by strong underlying surface changes; therefore, the impact of a reservoir on hydrological alteration is difficult to extract from these changes. The Taizi River, which has a relatively stable climate and underlying surface is a suitable site for this research. The Shenwo Reservoir (SWR) in the middle reaches can be considered the most important cause of the hydrological regime and ecosystem variation in this basin. Previous studies on the Taizi River have focused on the annual or seasonal changes in streamflow, with little attention being given to the hydrological alterations and their ecological impact [27,28]. Therefore, the objectives of this study were as follows: (1) to analyze the reservoir-induced surplus and deficit of eco-flow using two eco-flow indicators, the ecosurplus and ecodeficit; (2) evaluate the reservoir-induced hydrological alteration and its ecological impact; and (3) compare eco-flow indicators with IHAs.

## 2. Materials and Methods

### 2.1. Study Area

The Taizi River Basin (TRB), which is located in the Liaoning Province of China, originates from Changbai Mountain and has a length of 413 km. It flows through Benxi City, Liaoyang City, and Anshan City, with a drainage area of 13,883 km$^2$ (Figure 1). The TRB has a temperate monsoon climate, with an average annual precipitation of 700–900 mm that is mainly concentrated in July–September, which account for 70% of the annual precipitation. The seasonal division of the TRB is based on runoff, and the monthly and seasonal runoff distribution of Liaoyang Station before the construction of SWR is shown in Figure 2.

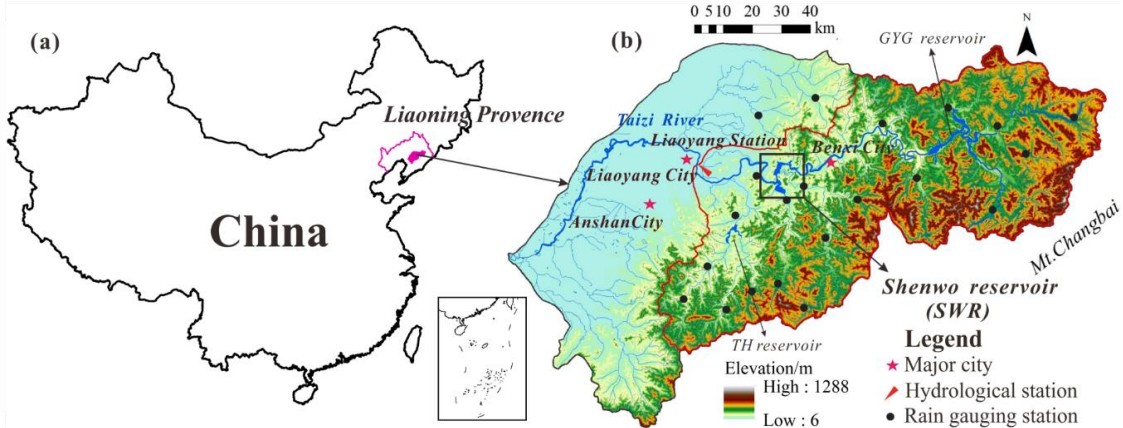

**Figure 1.** Location of: (**a**) the study area in China; and (**b**) hydrological station and rain gauging stations in TRB. The drainage area controlled by Liaoyang Station is surrounded by the red line, while the entire basin is surrounded by the black line.

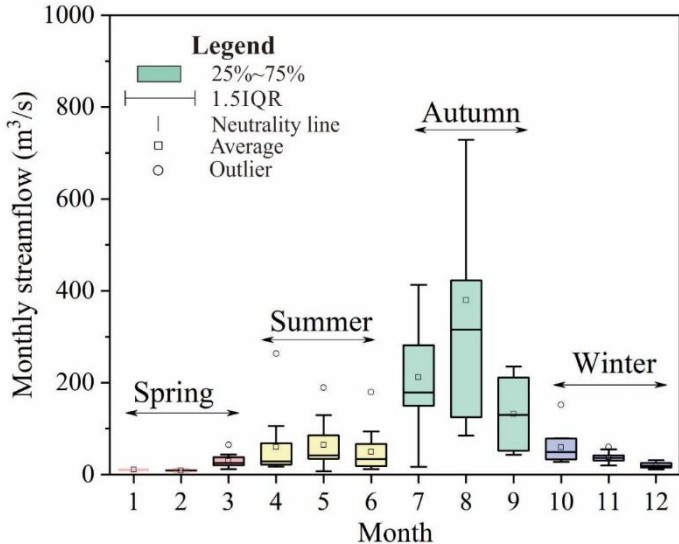

**Figure 2.** Monthly Streamflow distribution of Liaoyang station before the construction of SWR.

To reduce the threat of seasonal flooding and meet the water supply needs of industry, agriculture, and living, many reservoirs have been built in TRB. Three large reservoirs were built, including the Guanyinge (GYG) Reservoir (upstream of the main stream), the Shenwo Reservoir (midstream of the main stream), and the Tanghe (TH) Reservoir (tributary), which were completed in 1995, 1974, and 1969, respectively (Figure 1). The total storage capacity of the three reservoirs is 36.655 × 10$^8$ m$^3$, and their functions are mainly flood control, as well as secondary functions such as domestic and industrial

water supply, agricultural irrigation, power generation, and aquaculture. Table 1 gives information on these reservoirs in terms of construction time span, drainage area, total capacity, and average annual runoff. The annual average runoff of GYG Reservoir is 40.2% of SWR, and that of TH Reservoir is only 11.96%. Therefore, the regulation and storage of SWR plays a vital role in the ecosystem of the middle and lower reaches of the TRB.

**Table 1.** Information of major water reservoirs in Taizi River Basin.

| Names | Construction Duration | Drainage Area (km²) | Total Capacity ($10^8$ m³) | Average Annual Runoff ($10^8$ m³) |
|---|---|---|---|---|
| Guanyinge (GYG reservoir) | 1990–1995 | 2795 | 21.680 | 9.85 |
| Shenwo (SWR) | 1960–1974 | 3380 | 7.910 | 24.50 |
| Tanghe (TH reservoir) | 1958–1969 | 1228 | 7.065 | 2.93 |

*2.2. Data*

The lower reaches of the TRB are a typical agricultural plain with several water intakes along the river. Therefore, the Liaoyang Station in the middle reaches of the TRB was selected as the study station. In addition, the Liaoyang Station, which is located 25.5 km downstream, is the closest hydrological station to the SWR; therefore, the flow data can directly reflect the SWR regulation. The daily streamflow data from 1961 to 2016 were obtained from the Liaoning Provincial Department of Water Resources. This article focuses on the impact of the construction of SWR on the downstream hydrological regime alternation and ecological flow, and the streamflow data are divided into two series: 1961–1973 and 1974–2016. The monthly average precipitation data of 21 rain gauging stations (Figure 1) from 1961 to 2016 were obtained from the China Meteorological Data Network (data.cma.cn). The precipitation of the drainage area controlled by Liaoyang Station was calculated by the Thiessen Polygon method.

*2.3. Methods*

2.3.1. Ecosurplus and Ecodeficit

Two generalized indicators, ecosurplus and ecodeficit, based on the FDC, can reflect the overall gain or loss, respectively, in streamflow that results from flow regulation during a period [24]. The FDC illustrates the percentage of time ($P_i$) a given streamflow ($Q_i$) was equaled or exceeded during a specified period of time [29]. When $Q_i$ is arranged in descending order, $P_i$ can be expressed as

$$P_i = i/(n + 1) \tag{1}$$

where *i* is the rank of $Q_i$ and *n* is the sample size of $Q_i$. Since the introduction of ecosurplus and ecodeficit, various calculation methods have been developed [22,24,30]. As eco-flow varies within a certain range, this study defined the range as the upper and lower limits of eco-flow before the reservoir was constructed, which are different from the original definition [24]. First, construct the pre-reservoir annual or seasonal FDCs of each year; then, calculate the 25%, 50%, and 75% quantiles of $Q_i$ corresponding to each $P_i$. Then, construct 25%, 50%, and 75% of the FDC. In this study, 75% and 25% of the FDC were used as the upper and lower limits of eco-flow. The ecosurplus and ecodeficit are calculated as

$$\text{Ecosurplus} = A_s/A_{50\%} \tag{2}$$

$$\text{Ecodeficit} = A_d/A_{50\%} \tag{3}$$

where $A_s$ is the area enclosed by the 75% of the FDC and the portion of the FDC higher than 75% of the FDC in the specified year (or season); $A_d$ is the area enclosed by 25% of the FDC and the FDC lower than 25% of the FDC; and $A_{50\%}$ is the annual or seasonal flow corresponding to 50% FDC, as shown in Figure 3. Rather than being divided by the mean annual or seasonal flow [22,23], this calculation is

divided by the flow corresponding to 50% of the FDC (shaded part in Figure 3). As the median is less affected by extreme events relative to the mean, the ecological indicators can be better quantified [30].

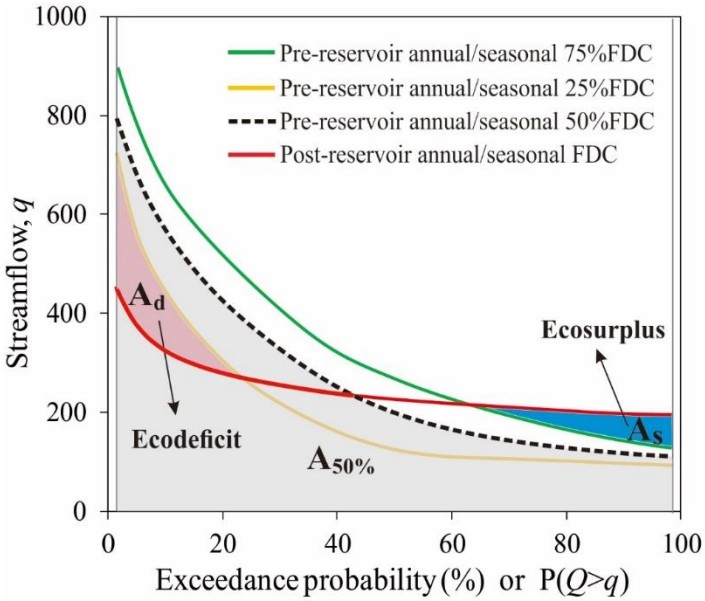

**Figure 3.** Schematic diagram of the definitions of ecosurplus and ecodeficit.

#### 2.3.2. Evaluation of Hydrological Alteration

The 33 IHAs were used to assess the specific impact of the reservoir on hydrological alterations. As there is no "zero-flow day" in the streamflow data before the SWR, the 32 IHAs are finally used, as shown in Table 2. Each indicator has its ecological relevance [31].

**Table 2.** Summary of the 32 IHAs.

| IHA Parameter Group | Hydrologic Parameters |
| --- | --- |
| 1. Magnitude of monthly water conditions | Mean value for each calendar month: Mean flow in January; mean flow in February; mean flow in March; mean flow in April; mean flow in May; mean flow in June; mean flow in July; mean flow in August; mean flow in September; mean flow in October; mean flow in November; mean flow in December |
| 2. Magnitude and duration of annual extreme water conditions | 1-day minimum; 3-day minimum; 7-day minimum; 30-day minimum; 90-day minimum; 1-day maximum; 3-day maximum; 7-day maximum; 30-day maximum; 90-day maximum; base flow index (7-day minimum flow/mean flow for year) |
| 3. Timing of annual extreme water conditions | Day of year of each annual 1-day maximum: date of minimum;Day of year of each annual 1-day minimum: date of maximum; |
| 4. Frequency and duration of high and low pulses | Number of low pulses within each water year: low pulse count;mean or median duration of low pulses (days): low pulse duration;number of high pulses within each water year: high pulse count;mean or median duration of high pulses (days) high pulse duration |
| 5. Rate and frequency of water condition changes | Mean of all positive differences between consecutive daily values: rise rates;mean of all negative differences between consecutive daily values: fall rates;number of hydrologic reversals: numbers of reversals |

The most ecologically relevant hydrological indicators were identified from the 32 IHAs by principal component analysis (PCA) to evaluate the hydrological alteration. According to the Kaiser–Guttman criterion, components with eigenvalues greater than 1 were retained [32], and the IHA parameter with the largest load was selected in each principal component (PC) to represent each PC.

2.3.3. Evaluation of Biodiversity

The Shannon Index (*SI*) is the most widely used indicator for assessing biodiversity [33]. The larger is the *SI*, the richer is the biodiversity [14,34]. The *SI* is expressed as:

$$SI = -\sum p_i \times \log p_i \tag{4}$$

where $p_i$ is the ratio of the ith species to the total number of species in a community. Yang determined a quantitative function between *SI* and the 32 IHAs based on genetic programming [10]:

$$SI = \frac{D_{\mathrm{min}}/Q_{\mathrm{7daymin}} + D_{\mathrm{min}}}{Q_3 + Q_5 + Q_{\mathrm{3daymin}} + 2 \cdot Q_{\mathrm{3daymax}}} + R_{\mathrm{rate}} \tag{5}$$

where $D_{\mathrm{min}}$ represents the date of minimum flow in a year; $Q_{\mathrm{7daymin}}$ and $Q_{\mathrm{3daymin}}$ represent the 7-day minimum and 3-day minimum flows, respectively; $Q_{\mathrm{3daymax}}$ represents the 3-day maximum flow; $Q_3$ and $Q_5$ represent the mean flow for March and May, respectively; and $R_{\mathrm{rate}}$ represents the rise rates. Although Equation (5) was originally established based on the data of fish community in the upper Illinois River, it has been applied to the assessment of the biodiversity of the East River [23], Yellow River [26], Lake Shelbyville [35], etc. Owing to the lack of data on the biodiversity in TRB, the biodiversity can be estimated roughly according to Equation (5). In fact, each river has its own ecological uniqueness, and the applicability of Equation (5) in TRB needs further verification by the measured data of the biological community.

## 3. Results and discussion

### 3.1. Variations in Eco-Flow Indicators

3.1.1. Changes in Flow Components from the Perspective of Eco-Flow Indicators

The construction of a reservoir will cause changes in the flow components of downstream rivers, such as high and low flow components. Figure 4 shows the pre-reservoir and post-reservoir FDC scatters in years and seasons. The values and frequencies of post-reservoir high-flow are smaller than those of the pre-reservoir high-flow, and the high-flow below 25% FDC will lead to an increase in the ecodeficit. The post-reservoir low-flow component is higher than that pre-reservoir low-flow, and the low-flow above 75% FDC will produce an ecosurplus. The changes in flow components are closely related to the seasonal regulation of the reservoir. As the streamflow of autumn accounts for the largest proportion of annual streamflow (Figure 2), the change in high-flow after construction of the SWR (the decreases in value and frequency) is mainly reflected in the autumn, while the low-flow component can better cover the areas where low-flow occurs before the reservoir, which inevitably leads to an increase in the ecodeficit in autumn. The post-reservoir FDC changes in spring and winter are similar, showing a situation where the peak is shifted down, while the tail is moved up. Furthermore, the difference between high and low flows in most years reduced. The values and frequency of the post-reservoir summer high-flow and low-flow component are higher than those of the pre-reservoir flow component, which is related to the water demand for irrigation in the downstream paddy fields in summer and will inevitably lead to an increase in the ecosurplus and a reduction in the ecodeficit. The change in the FDC scatters plot can preliminarily determine the impact of the reservoir on eco-flow indicators. However, the variations in the eco-flow indicators are also closely related to precipitation anomalies.

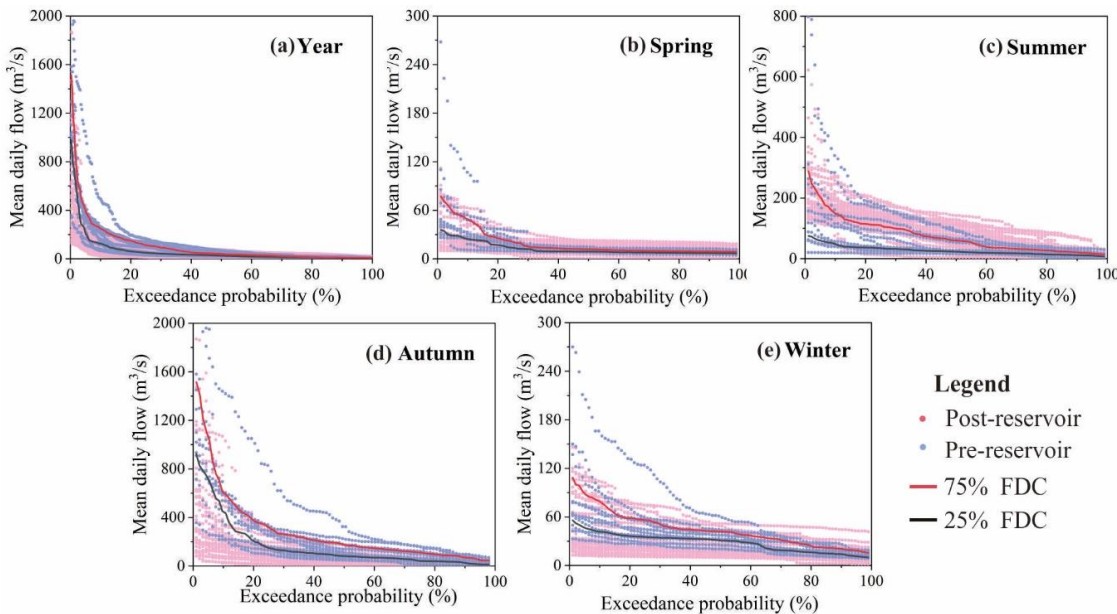

**Figure 4.** Annual and seasonal FDC scatter plots before and after the construction of the reservoir: (**a**) year; (**b**) spring; (**c**) summer; (**d**) autumn; and (**e**) winter.

### 3.1.2. Relationship between Eco-Flow Indicators and Precipitation

Figure 5 shows the annual and seasonal eco-flow indicator variations and precipitation anomalies before and after construction of the SWR, and Table 3 shows the correlation coefficient between the two. From the annual scale, the changes in the ecosurplus and precipitation are relatively consistent in the entire period (correlation coefficients are 0.76 and 0.80, respectively), especially in years when a considerable amount of rain is received. As the precipitation increases the high-flow component of the annual runoff, the part of the FDC exceeding 75% of the FDC increased, leading to increased ecosurplus. However, it was observed that the extremely rainy years following the construction of the SWR could not produce a correspondingly large ecosurplus, such as 1985, 1995, and 2010, which indicates that, as the precipitation increases to a certain extent, the resulting ecosurplus is limited by the reservoir and reaches a certain upper limit, which is between 0.9 and 1.3. On the other hand, as the demand for water supply in TRB increases year by year, part of the ecosurplus generated by precipitation is used to supply industry and domestic water. In successive dry years, such as 1976–1984 and 1996–2009, the ecosurplus is basically zero, and the rainy year following the successive dry years shows a relatively small ecosurplus, which is mainly affected by reservoir impoundment. Compared with the ecosurplus, because the overall drought that is experienced after the construction of the reservoir leads to an increase in the fraction below 25% FDC, the post-reservoir ecodeficit is higher than the pre-reservoir ecodeficit. The pre-reservoir ecodeficit was well correlated with precipitation. However, affected by the reservoir, the correlation becomes worse, especially in years during which a drought was experienced, such as 1979, 1989, and 2014, as the ecodeficit is not proportional to the drought regime. This is because the opening and discharging of water by the reservoir to meet the downstream water demand increased the ecodeficit.

Compared with the variation on the annual scale, the difference between the two increases on the seasonal scale. The changes in autumn, during which the most precipitation is experienced, were most similar to the changes on the annual scale. The eco-flow indicators and precipitation were in good agreement (Table 3). However, the consistency of the eco-flow indicators and precipitation in spring, summer, and winter is poor, with correlation coefficients that are less than 0.6, indicating that the ecosurplus and ecodeficit are significantly affected by the construction of the SWR. In addition, except for the ecosurplus in autumn, the correlation coefficients (absolute values) between the remaining seasonal eco-flow indicators and precipitation have decreased to varying degrees after the SWR (Table 3),

which indicates the impact of reservoir regulation on eco-flow indicators. For autumn, precipitation still plays a leading role in ecosurplus. In spring and summer, the river is mainly characterized by ecosurplus, even in the extreme dry or successive dry period. A significantly decreased ecodeficit was observed after the construction of the reservoir, reflecting the role of the reservoir in maintaining the low flow in spring and especially in summer (Figure 4). In winter, the trends of the ecosurplus and ecodeficit are similar to those in autumn, and to some extent inherit the characteristics of eco-flow indicators in autumn. As the temperature in winter is so low that snowfall cannot form a runoff, the correlation between the eco-flow indicators and precipitation is poor. The eco-flow indicators in winter may be controlled by autumn precipitation and reservoir regulation.

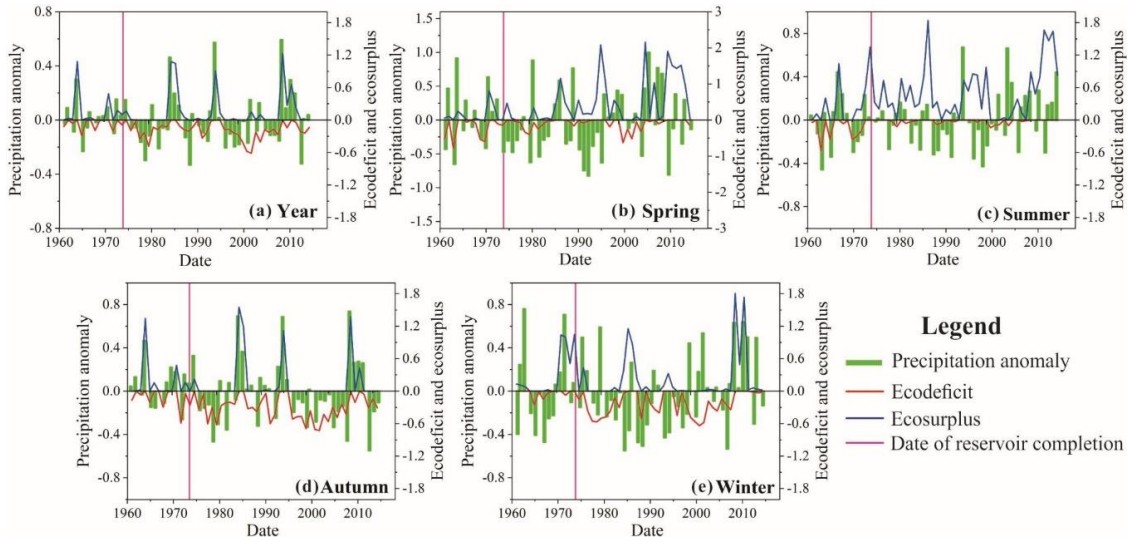

**Figure 5.** Variations in eco-flow indicators and precipitation anomalies: (**a**) year; (**b**) spring; (**c**) summer; (**d**) autumn; and (**e**) winter.

**Table 3.** Correlation coefficient between eco-flow indicators and precipitation before and after the SWR.

| Correlation Coefficient | Annual | | Spring | | Summer | | Autumn | | Winter | |
|---|---|---|---|---|---|---|---|---|---|---|
| | Eoc-s [1] | Ecod [1] | Ecos | Ecod | Ecos | Ecod | Ecos | Ecod | Ecos | Ecod |
| Before SWR | 0.76 | −0.68 | 0.53 | −0.24 | 0.60 | −0.52 | 0.70 | −0.66 | 0.50 | −0.34 |
| After SWR | 0.80 | −0.44 | 0.17 | −0.04 | 0.10 | −0.08 | 0.79 | −0.61 | 0.25 | −0.10 |

[1] Ecos and Ecod represent Ecosurplus and Ecodeficit, respectively.

### 3.2. Hydrological Regime Alteration Using the IHA Method

Table 4 lists the changes in 32 IHAs before and after the construction of the reservoir, with 24 IHAs exhibiting a relative change rate of more than 20%, indicating that the reservoir has a greater impact on the hydrological regime. The average monthly flow in spring and summer is generally increased, which is consistent with the upward trend of the seasonal FDC in Figure 4 and the increase in the ecosurplus in Figure 5. The streamflow in autumn showed a decreasing trend, which was consistent with the downward trend of the autumn FDC in Figure 4 and the increase in the ecodeficit in Figure 5. Even though the post-reservoir climate tends to be arid, under the regulation of the SWR, the streamflow during the dry season (spring and winter) still increased from 8.89–22.81 to 19.49–24.41 m$^3$/s, which increased the 1-, 3-, 7-, and 30-day minimum flows, leading to an increase in the low-flow component value, which is consistent with the change in the spring FDC in Figure 4. On the other hand, the regulation of the SWR significantly reduced the flow fluctuation during the wet season, from 116.86–276.87 to 120.62–148.98 m$^3$/s, which reduced the 1-, 3-, 7-, 30- and 90-day maximum flows significantly, leading to a decrease in the high-flow component, which is consistent

with the change in the autumn FDC in Figure 4. The base flow index (BFI) increased by 45%, but this was not due to an increase in actual base flow; it was the result of reservoir regulation. The date of maximum flow did not change significantly, but the date of minimum flow was postponed by 22%. The low and high pulse counts increased, while the duration decreased, which is closely related to the hydropower generation of the SWR [36] and stabilized the downstream hydrological processes to a certain extent. Significant changes in the rise and fall rates and the number of reversals were not observed. Overall, the changes in the 32 IHAs coincide with the changes in the FDC and eco-flow indicators, which gives us more confidence using the ecosurplus and ecodeficit to evaluate ecological flow. The relationship between the 32 IHAs and the ecosurplus and ecodeficit is discussed below.

**Table 4.** IHAs and their relative changes before and after the construction of the SWR.

| IHA | Units | Pre-Reservoir (1961–1974) | Post-Reservoir (1975–2017) | Relative Change (%) |
|---|---|---|---|---|
| Mean flow in January | $m^3/s$ | 10.64 | 20.40 | **0.92** [1] |
| Mean flow in February | $m^3/s$ | 8.89 | 19.49 | **1.19** |
| Mean flow in March | $m^3/s$ | 22.81 | 21.39 | −0.06 |
| Mean flow in April | $m^3/s$ | 47.18 | 28.41 | **−0.40** |
| Mean flow in May | $m^3/s$ | 50.81 | 123.57 | **1.43** |
| Mean flow in June | $m^3/s$ | 41.97 | 83.69 | **0.99** |
| Mean flow in July | $m^3/s$ | 116.86 | 120.62 | 0.03 |
| Mean flow in August | $m^3/s$ | 276.87 | 148.98 | **−0.46** |
| Mean flow in September | $m^3/s$ | 100.88 | 58.99 | **−0.42** |
| Mean flow in October | $m^3/s$ | 55.53 | 40.43 | **−0.27** |
| Mean flow in November | $m^3/s$ | 37.82 | 31.18 | −0.18 |
| Mean flow in December | $m^3/s$ | 18.95 | 24.41 | **0.29** |
| 1-day minimum | $m^3/s$ | 6.83 | 8.29 | **0.21** |
| 3-day minimum | $m^3/s$ | 7.07 | 8.66 | **0.22** |
| 7-day minimum | $m^3/s$ | 7.46 | 9.53 | **0.28** |
| 30-day minimum | $m^3/s$ | 8.38 | 11.97 | **0.43** |
| 90-day minimum | $m^3/s$ | 16.53 | 16.97 | 0.03 |
| 1-day maximum | $m^3/s$ | 1455.91 | 787.36 | **−0.46** |
| 3-day maximum | $m^3/s$ | 1265.55 | 713.57 | **−0.44** |
| 7-day maximum | $m^3/s$ | 948.76 | 570.27 | **−0.40** |
| 30-day maximum | $m^3/s$ | 477.75 | 311.08 | **−0.35** |
| 90-day maximum | $m^3/s$ | 255.91 | 178.88 | **−0.30** |
| Base flow index | $m^3/s$ | 0.10 | 0.14 | **0.45** |
| Date of minimum | day | 102.55 | 125.13 | **0.22** |
| Date of maximum | day | 203.73 | 202.62 | −0.01 |
| Low pulse count | - | 3.82 | 6.16 | **0.61** |
| Low pulse duration | day | 14.50 | 9.70 | **−0.33** |
| High pulse count | - | 4.09 | 7.38 | **0.80** |
| High pulse duration | day | 15.64 | 10.88 | **−0.30** |
| Rise rate | $m^3/s/day$ | 2.17 | 2.41 | 0.11 |
| Fall rate | $m^3/s/day$ | −2.31 | −2.12 | −0.08 |
| Numbers of reversals | - | 99.64 | 111.20 | 0.12 |

[1] Change rates greater than 20% are highlighted in bold.

*3.3. Impact of Hydrological Alteration on Downstream Biodiversity*

3.3.1. Ecologically Relevant Hydrological Indicators

Similar to other hydrological indicators, IHAs are interrelated [15]. Figure 6 is a boxplot of the correlation coefficient between each IHA parameter and the remaining 31 IHAs. The absolute value of the correlation coefficient is between 0.003 and 0.995, with an average value of 0.33. It should be noted that some values are even larger than 0.9, which illustrates the statistical redundancy of the 32 IHAs. Therefore, it is important to determine a small set of ecologically relevant hydrological indicators

(ERHI) to characterize changes in the eco-flow regime. The results from the PCA method are shown in Table 5. Six principal components (PC) are selected, which are the 30-day minimum, BFI, mean flow in May, high pulse duration, number of reversals, and date of minimum. The selected six ERHIs are distributed in five IHA groups (Table 2); therefore, the ERHIs can reflect the five characteristics of hydrological regime and are poorly related to each other. The results are similar to previous research results [10,23,26].

**Table 5.** ERHI results using the principal component analysis method.

| IHAs | PC1 | PC2 | PC3 | PC4 | PC5 | PC6 |
|---|---|---|---|---|---|---|
| Mean flow in January | 0.66 | 0.62 | 0.23 | 0.02 | −0.13 | 0.07 |
| Mean flow in February | 0.60 | 0.58 | 0.33 | 0.09 | 0.00 | 0.12 |
| Mean flow in March | 0.65 | 0.20 | 0.17 | 0.16 | −0.46 | 0.30 |
| Mean flow in April | 0.36 | −0.27 | −0.51 | 0.03 | −0.39 | 0.46 |
| Mean flow in May | 0.38 | 0.36 | **−0.66** [1] | 0.30 | 0.56 | 0.12 |
| Mean flow in June | 0.46 | 0.21 | −0.54 | 0.56 | 0.27 | 0.12 |
| Mean flow in July | 0.65 | −0.36 | −0.11 | 0.30 | 0.20 | −0.11 |
| Mean flow in August | 0.60 | −0.61 | 0.21 | 0.00 | −0.03 | 0.16 |
| Mean flow in September | 0.67 | −0.50 | 0.09 | −0.16 | 0.09 | −0.14 |
| Mean flow in October | 0.67 | −0.16 | −0.15 | −0.28 | −0.08 | −0.07 |
| Mean flow in November | 0.60 | 0.04 | −0.47 | −0.25 | −0.11 | −0.31 |
| Mean flow in December | 0.60 | 0.23 | −0.22 | −0.22 | 0.10 | −0.33 |
| 1-day minimum | 0.82 | 0.44 | 0.18 | 0.06 | −0.05 | −0.14 |
| 3-day minimum | 0.82 | 0.45 | 0.20 | 0.04 | −0.04 | −0.12 |
| 7-day minimum | 0.79 | 0.53 | 0.21 | 0.02 | −0.05 | −0.07 |
| 30-day minimum | **0.85** | 0.59 | 0.20 | −0.01 | −0.05 | −0.08 |
| 90-day minimum | 0.74 | 0.30 | 0.19 | −0.05 | −0.24 | 0.04 |
| 1-day maximum | 0.67 | −0.63 | 0.24 | −0.12 | 0.16 | 0.05 |
| 3-day maximum | 0.68 | −0.61 | 0.23 | −0.14 | 0.17 | 0.04 |
| 7-day maximum | 0.70 | −0.59 | 0.20 | −0.11 | 0.15 | 0.06 |
| 30-day maximum | 0.72 | −0.61 | 0.18 | −0.01 | 0.13 | 0.13 |
| 90-day maximum | 0.75 | −0.60 | 0.12 | 0.06 | 0.13 | 0.07 |
| Base flow index | 0.16 | **0.84** | 0.23 | −0.01 | −0.03 | −0.03 |
| Date of minimum | −0.24 | 0.34 | 0.37 | −0.21 | 0.09 | **0.56** |
| Date of maximum | 0.12 | −0.37 | 0.05 | −0.38 | −0.10 | 0.13 |
| Low pulse count | −0.73 | 0.15 | 0.22 | −0.28 | 0.09 | 0.03 |
| Low pulse duration | −0.18 | −0.26 | −0.21 | 0.63 | −0.21 | 0.22 |
| High pulse count | 0.22 | 0.47 | −0.36 | −0.28 | 0.13 | 0.10 |
| High pulse duration | 0.07 | −0.27 | 0.13 | **0.74** | −0.10 | −0.31 |
| Rise rate | 0.69 | 0.09 | −0.58 | −0.06 | −0.06 | 0.17 |
| Fall rate | −0.77 | 0.00 | 0.56 | 0.05 | 0.11 | 0.00 |
| Number of reversals | 0.12 | 0.50 | −0.14 | −0.03 | **0.61** | 0.28 |

[1] Selected principal components are highlighted in bold.

The variations in the ERHIs over time are shown in Figure 7, and the trend is fitted by locally weighted regression (Loess). After the construction of the SWR, the mean flow in May increased significantly, with an average increase of 143% (Table 4) and remained high. May is the migratory and spawning period of the fish in the TRB. Dugan and Larinier stated that, when the flow rate of the river exceeds the natural condition, the migration distance and velocity will decrease as the flow rate increases, and migration difficulties will occur [37,38]. Obviously, the increase in flow in May will have an adverse impact on the migration and reproduction of fish in the TRB. The 30-day minimum flow oscillating amplitude significantly increased after 1985, and the BFI showed an increasing trend, with a change range that also increased. The change in minimum flow will rebalance the competitive, ruderal, and stress-tolerant organisms. The minimum flow before the construction of the SWR occurred mainly in the spring, and in winter and summer after the construction of the SWR. In general, the minimum flow day presents a postponement, which will have a potential impact on river ecology, especially the

reproduction of aquatic organisms [26]. After the construction of the SWR, the high pulse duration was significantly shorter and remained at a lower level, with an average reduction of 30% (Table 3), which directly influences the bedload transport, channel sediment textures, and duration of substrate disturbance, and reduces the time for nutrient and organic matter exchanges between the river and floodplain. The number of reversals slightly increased, which may cause slight stress on low-mobility streamedge organisms. Changes in ERHIs after the construction of the SWR will cause changes in the ecosurplus and ecodeficit, leading to changes in the river ecosystem [23].

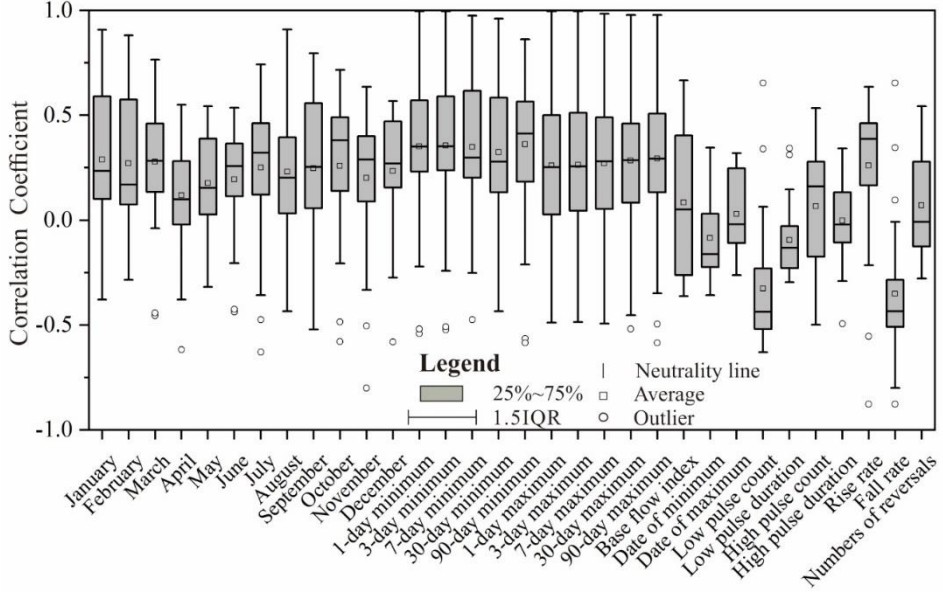

**Figure 6.** Boxplot of the correlation coefficients between the 32 IHAs.

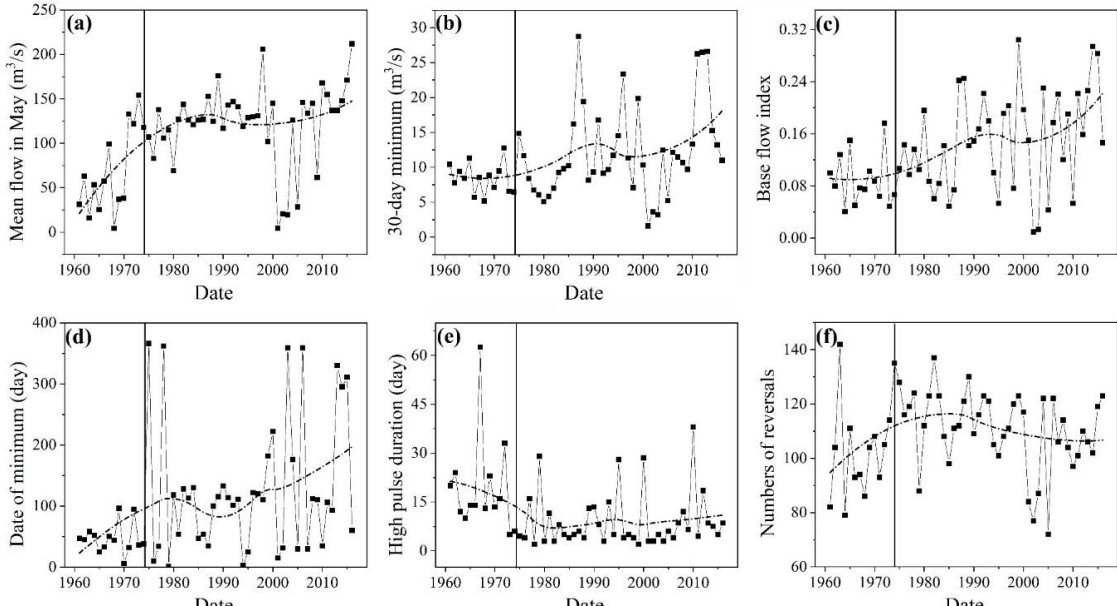

**Figure 7.** Temporal variation in the ERHIs: (**a**) mean flow in May; (**b**) 30-day minimum; (**c**) base flow index; (**d**) date of minimum; (**e**) high pulse duration; and (**f**) number of reversals. The dotted line is the Loess function fitting curve, and the black vertical line is the date when the SWR completed.

In addition, it can be observed that ERHIs changed in 1995 (Figure 7), which may be caused by the impoundment of newly built GYG Reservoir upstream. It should be noted that the total capacity of GYG Reservoir is about three times that of SWR (Table 1). Taking the average flow in May as an

example, before the GYG Reservoir, the flow was maintained within 70–206 m$^3$/day. However, after the GYG Reservoir, there was a situation below 30 m$^3$/day between 2001 and 2005, which was closely related to the runoff interception by GYG Reservoir. On the other hand, the climate tended to be arid after 1995, and it did not moderate until 2010, which also had a certain impact on the change of ERHIs. Similarly, in Figures 5 and 8 (mentioned in Section 3.3.2), similar changes are also observed.

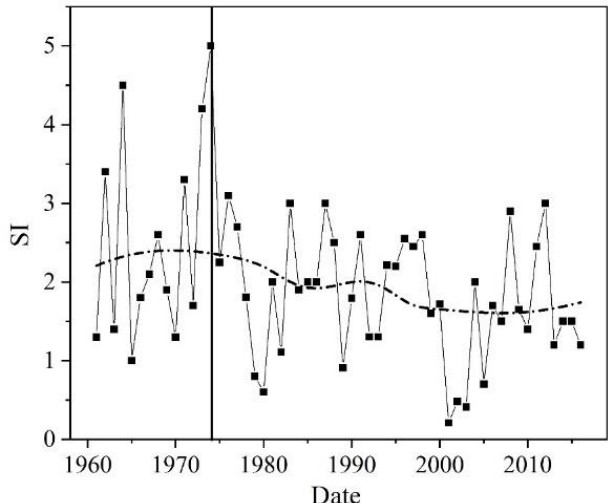

**Figure 8.** Temporal variation in *SI*. The dotted line is the Loess function fitting curve.

### 3.3.2. Ecological Diversity Indicator: SI

Figure 8 shows the temporal variation in the ecological diversity index *SI*, and regression fitting was performed by the Loess function. After the SWR was constructed, the *SI* was generally smaller, indicating the negative impact of the SWR on the downstream river ecosystem. In the natural situation, changes in precipitation directly control changes in flow, indirectly affecting biodiversity. Figure 9 shows the relationship between precipitation and *SI* before and after the SWR was constructed. A good correlation between precipitation and *SI* was observed, with $R^2 = 0.79$, and *SI* increased with an increase in precipitation. However, the correlation decreased after the SWR was constructed, with $R^2 = 0.042$, indicating that the increase in precipitation was not enough to cause an increase in ecological diversity. This also highlights the negative impact of the construction of the SWR on the ecological changes of downstream rivers. As indicated in Section 3.3.1, the SWR can affect the downstream ecology by changing the hydrological regime, but the changes in the hydrological regime are not the only explanation. On the one hand, the SWR blocks the migratory channels of migratory fish and affects the interspecific gene flow of the entire system. On the other hand, the water discharged from the reservoir originates from the deep part of the reservoir. In summer, the temperature is lower than that of the river, while, in winter, it is higher than that in the river. This will change the living environment and life cycle of aquatic organisms, as the processes of reproduction and hatching depend on the temperature change. In addition, a large amount of gravel is intercepted, which causes invertebrates at the bottom of the riverbed, such as mollusks and shellfish, to lose their living environment. Although the *SI* decreased after the SWR was constructed, the change range was smaller than that before the SWR was constructed. After the completion of the GYG Reservoir (1995), a new equilibrium state appears in the downstream river ecosystem, where the ecological diversity was less affected by precipitation and mainly controlled by the regulation of the SWR.

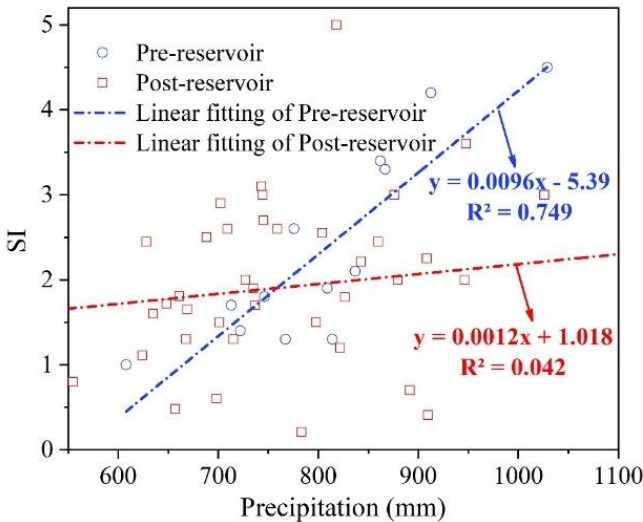

**Figure 9.** Relationship between *SI* and precipitation.

*3.4. Comparison between Ecological Indicators and IHA Indicators*

To understand the relationship between the eco-flow indicators and 32 IHAs, the correlation coefficients between them were calculated, as shown in Table 6. It can be seen that there is a good correlation between most eco-flow indicators and the 32 IHAs. The ecosurplus is positively correlated with most of the 32 IHAs, while the ecodeficit is negatively correlated with the IHAs. Each eco-flow indicator is significantly correlated with 9–21 IHAs, and most IHAs are also significantly correlated with at least one ecological flow indicator. In this respect, eco-flow indicators have a smaller statistical redundancy than the 32 IHAs. The monthly mean flow of each season is highly correlated with the corresponding seasonal eco-flow indicators, which indicates that the seasonal eco-flow index can reflect the variation in monthly streamflow. In addition, the correlation between extreme flow and eco-flow indicators is obvious. The minimum flow is most strongly correlated with the spring ecosurplus, while the maximum flow is strongly correlated both with the ecosurplus in autumn and the year. As the minimum flow occurs mostly in spring, the BFI is related to the spring eco-flow index. The rise rate is positively correlated with the ecosurplus and negatively correlated with the ecodeficit, while the fall rate exhibits an opposite trend.

**Table 6.** Correlation coefficient between the 32 IHAs and eco-flow indicators.

| 32 IHAs | Spring | | Summer | | Autumn | | Winter | | Year | |
|---|---|---|---|---|---|---|---|---|---|---|
| | Ecos [1] | Ecod | Ecos | Ecod | Ecos | Ecod | Ecos | Ecod | Ecos | Ecod |
| Mean flow in January | **0.77** [2] | **−0.41** | 0.20 | −0.28 | −0.03 | −0.11 | 0.17 | −0.16 | 0.10 | −0.16 |
| Mean flow in February | **0.80** | −0.29 | 0.21 | −0.22 | −0.04 | −0.15 | 0.15 | −0.25 | 0.13 | −0.17 |
| Mean flow in March | **0.65** | **−0.58** | **0.35** | **−0.38** | 0.10 | −0.22 | 0.09 | −0.31 | 0.17 | **−0.35** |
| Mean flow in April | 0.04 | −0.25 | **0.41** | −0.30 | 0.27 | −0.25 | 0.16 | −0.22 | 0.30 | −0.32 |
| Mean flow in May | 0.32 | **−0.39** | **0.49** | **−0.55** | 0.09 | −0.06 | 0.25 | −0.15 | 0.15 | −0.28 |
| Mean flow in June | 0.20 | −0.29 | **0.54** | **−0.42** | 0.26 | −0.26 | **0.42** | −0.19 | **0.38** | **−0.39** |
| Mean flow in July | 0.07 | **−0.35** | 0.29 | **−0.35** | **0.52** | **−0.64** | 0.29 | **−0.38** | **0.55** | **−0.64** |
| Mean flow in August | −0.04 | −0.29 | 0.18 | −0.25 | **0.91** | **−0.60** | **0.47** | **−0.41** | **0.90** | **−0.55** |
| Mean flow in September | 0.06 | −0.32 | 0.06 | −0.31 | **0.74** | **−0.66** | **0.63** | **−0.59** | **0.79** | **−0.63** |
| Mean flow in October | 0.06 | **−0.35** | 0.15 | −0.28 | **0.52** | **−0.43** | **0.84** | **−0.59** | **0.61** | **−0.52** |
| Mean flow in November | 0.08 | −0.33 | 0.22 | −0.24 | **0.35** | **−0.46** | **0.75** | **−0.65** | **0.47** | **−0.51** |
| Mean flow in December | 0.34 | −0.31 | 0.21 | −0.25 | 0.28 | −0.30 | **0.63** | **−0.50** | **0.44** | **−0.35** |
| 1-day minimum | **0.68** | **−0.60** | **0.43** | **−0.44** | 0.13 | **−0.40** | **0.40** | **−0.50** | 0.30 | **−0.46** |
| 3-day minimum | **0.68** | **−0.60** | **0.43** | **−0.44** | 0.13 | **−0.39** | **0.40** | **−0.50** | 0.30 | **−0.46** |
| 7-day minimum | **0.65** | **−0.61** | **0.40** | **−0.45** | 0.10 | **−0.36** | **0.38** | **−0.50** | 0.27 | **−0.45** |
| 30-day minimum | **0.64** | **−0.56** | **0.36** | **−0.43** | 0.09 | −0.27 | **0.38** | **−0.46** | 0.26 | **−0.37** |
| 90-day minimum | **0.69** | **−0.55** | 0.23 | **−0.40** | 0.16 | **−0.37** | **0.39** | **−0.50** | 0.33 | **−0.44** |
| 1-day maximum | −0.05 | −0.25 | 0.05 | −0.11 | **0.78** | **−0.73** | **0.43** | **−0.42** | **0.78** | **−0.65** |

**Table 6.** *Cont.*

| 32 IHAs | Spring | | Summer | | Autumn | | Winter | | Year | |
|---|---|---|---|---|---|---|---|---|---|---|
| | Ecos [1] | Ecod | Ecos | Ecod | Ecos | Ecod | Ecos | Ecod | Ecos | Ecod |
| 3-day maximum | −0.04 | −0.25 | 0.04 | −0.10 | **0.76** | **−0.72** | 0.45 | −0.42 | **0.77** | **−0.65** |
| 7-day maximum | −0.05 | −0.26 | 0.08 | −0.10 | **0.77** | **−0.72** | 0.47 | −0.43 | **0.79** | **−0.65** |
| 30-day maximum | −0.05 | −0.31 | 0.18 | −0.21 | **0.90** | **−0.72** | 0.46 | −0.47 | **0.89** | **−0.68** |
| 90-day maximum | −0.01 | **−0.35** | 0.23 | −0.29 | **0.90** | **−0.74** | 0.50 | −0.49 | **0.90** | **−0.72** |
| Base flow index | **0.48** | **−0.44** | 0.21 | −0.32 | **−0.37** | 0.23 | −0.11 | −0.09 | −0.30 | 0.09 |
| Date of minimum | 0.30 | −0.11 | 0.07 | 0.01 | −0.24 | 0.23 | −0.25 | 0.18 | −0.24 | 0.25 |
| Date of maximum | 0.11 | −0.16 | −0.09 | −0.09 | 0.23 | −0.25 | 0.24 | −0.26 | 0.21 | −0.30 |
| Low pulse count | −0.31 | **0.39** | −0.34 | **0.36** | −0.29 | **0.65** | −0.34 | **0.61** | **−0.38** | **0.73** |
| Low pulse duration | −0.22 | 0.12 | 0.12 | 0.04 | 0.02 | −0.04 | −0.15 | 0.25 | −0.02 | −0.01 |
| High pulse count | 0.24 | −0.11 | 0.27 | −0.24 | −0.21 | 0.13 | 0.20 | −0.25 | −0.12 | −0.04 |
| High pulse duration | −0.10 | −0.08 | 0.19 | −0.16 | 0.39 | −0.27 | 0.38 | −0.06 | **0.41** | −0.27 |
| Rise rate | 0.12 | **−0.39** | 0.47 | −0.43 | 0.34 | −0.43 | 0.60 | −0.53 | 0.44 | −0.56 |
| Fall rate | −0.24 | 0.45 | **−0.51** | 0.45 | −0.33 | 0.52 | **−0.56** | 0.67 | −0.46 | 0.63 |
| Numbers of reversals | −0.02 | −0.31 | 0.10 | −0.15 | −0.26 | 0.01 | −0.03 | −0.01 | −0.25 | −0.17 |

[1] Ecos and Ecod represent Ecosurplus and Ecodeficit, respectively. [2] Those at the 0.01 significant level are highlighted in bold.

These correlations indicate that eco-flow indicators can reflect information on changes in hydrological alteration on smaller time scales. As the eco-flow indicator is based on the annual or seasonal FDC, and the FDC cannot reflect the duration, time, and variation in a specific runoff event, the correlations between the eco-flow indicator and date of minimum, date of maximum, count, and duration of low and high pulses, and the numbers of reversals are quite weak.

In general, the eco-flow indicators can reflect most of the IHA information and its changing characteristics; therefore, the eco-flow indicators can provide a good evaluation standard for hydrological regime changes. In addition, the calculation of the ecosurplus and ecodeficit is based on the FDC and independent of the IHA, and, when using eco-flow indicators, there are no interrelated problems between indicators which widely exist in the IHA indicator system (Figure 6). Therefore, ecosurplus and ecodeficit can sufficiently handle the redundancy and correlations between hydrological indicators and can be used as indicators for the evaluation of hydrological alterations.

### 3.5. Suggestions Regarding the Operation of Reservoirs to Protect River Ecology

The high-flow components and frequencies are significantly reduced after the SWR, especially in autumn and winter, and the high-flow events correspond to the hydrological regime most needed by river ecosystems. Since the reservoir has functions such as flood control, water supply, and power generation, it is impractical to completely imitate the flow under natural conditions. Therefore, the reservoir manager should appropriately increase the high-flow and duration in autumn and winter according to the frequency of high-flow events before the SWR and the difficulty of implementation under the current conditions. In addition, the SWR caused an increase in the number of zero-flow days. The number of zero-flow days in 1990 was as much as 170 days, and the average annual zero-flow days during 1999–2004 was 45 days. Zero-flow days occurred mostly in continuous period, which definitely had a serious and irreversible impact on the river ecology. Therefore, it is recommended to further control the utilization of water resources in the TRB, especially the reasonable allocation of downstream irrigation and high-water-consuming industries. To ensure that there is no zero-flow in the downstream, it is essential to formulate strict water resources management regulations.

At the same time, due to the increase of low-flow components, the spring and summer mostly exhibited ecological surplus. Under the premise of ensuring downstream irrigation, the low flow rates in the spring and summer should be appropriately reduced. Controlling the discharge of the reservoir with reference to the natural hydrological regime of the river is an important means of protecting and repairing the river ecosystem. Therefore, it is recommended to use the median monthly flow before the SWR as the base flow. When the downstream flow is higher than the threshold, water should be

stored appropriately to adjust the high-flow in autumn and winter; when it is lower than this threshold, the reservoir impoundment and diversion should be stopped to maintain the ecological flow.

## 4. Conclusions

Based on the daily flow data from 1961 to 2016 in Liaoyang Station on the lower reaches of the SWR, the variations in ecosurplus and ecodeficit were analyzed. Together with the 32 IHAs, the ERHIs, and *SI*, the results were used to evaluate the impact of the construction of SWR on downstream hydrological alteration and ecological diversity. The main findings are as follows:

(1) Eco-flow indicators (ecosurplus and ecodeficit) can be used to analyze the annual and seasonal eco-flow variation due to the construction of the reservoir. After SWR was constructed, the high-flow value and frequency decreased, especially in autumn, which made the high-flow lower than 25% FDC, resulting in an ecodeficit; the post-reservoir low-flow value significantly increased, making the low-flow above 75% FDC and producing ecosurplus, especially in spring and summer.

(2) The relationship between the eco-flow indicators and precipitation can reflect the main factors affecting the eco-flow indicators. The annual ecosurplus is less affected by the reservoir, while the ecodeficit is greatly affected by the reservoir. Moreover, the seasonal ecological indicators are greatly affected by the reservoir, especially in spring and summer.

(3) After the SWR was constructed, the 32 IHAs showed significant changes, which were consistent with the changes in the ecosurplus and ecodeficit. Six ERHIs were screened by PCA, and their changes reduced the downstream ecological diversity. The ecological diversity was mainly controlled by reservoir regulation, and a new equilibrium state appeared.

(4) The eco-flow indicators have a good correlation with the 32 IHAs and can reflect the change information of most IHAs. Therefore, the eco-flow indicators can reflect the changing characteristics of the hydrological regime under the influence of the reservoir and provide a good evaluation standard. At the same time, the calculation of the ecosurplus and ecodeficit is independent of IHA, thus avoiding statistical redundancy.

**Author Contributions:** Conceptualization, M.L.; methodology, M.L. and X.L.; software, M.L. and X.Z.; formal analysis, M.L. and C.X.; investigation, M.L.; resources, G.L., H.L., and W.J.; writing—original draft preparation, M.L.; review and editing, X.L.; visualization, M.L.; supervision, C.X.; project administration, X.L. and C.X.; and funding acquisition, C.X. All authors have read and agreed to the published version of the manuscript.

**Funding:** The research was funded by Natural Science Foundation of China (No. 41572216), the China Geological Survey Shenyang Geological Survey Center "Hydrogeological investigation in the Songnen Plain" project ([2019]DD20190340-W09), the Provincial School Co-construction Project Special—Leading Technology Guide (SXGJQY2017-6), and the Jilin Province Natural Science Foundation (20140101164JC).This work was also partially funded by the Engineering Research Center of Geothermal Resources Development Technology and Equipment, Ministry of Education, Jilin University, China.

**Acknowledgments:** Special thanks are given to Professor Cao for her comments and concerns!

**Conflicts of Interest:** The authors declare no conflict of interest.

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
