# Peer review of "Evaluation of Reservoir-Induced Hydrological Alterations and Ecological Flow Based on Multi-Indicators"

_water, doi:10.3390/w12072069_

Round 1

Reviewer 1 Report

My comments below:

The detail of identifying the ecological impact is not clear. the ecological impact uncertainty is way less than the error introduced from other model parameters.

The method employed in this paper (Index method) -section 2.3.3 might be limited to specific site conditions. No-linearity among variables and parameters has not been well appreciated in the paper.

It would be advisable to consider another analytical and numerical modeling to complement or justify the conclusions arrived. Please see the suggested references below to consider alternative methods employed (for hydrology regime alteration) elsewhere.

Also, the time to reach equilibrium (system) to attain a steady-state after impacts) needs more clarification. Moreover, direct, indirect, and cumulative impacts are not explicitly mentioned either, perhaps due to the simplistic method applied.

Recommended References that can enrich the manuscript substantially includes:

Yihdego, Y and Webb, J.A., 2016. “Validation of a model with climatic and flow scenario analysis: case of Lake Burrumbeet in southeastern Australia” Journal of Environmental Monitoring & Assessment, 188, Article 308: 1-14. doi:10.1007/s10661-016-5310-7. http://link.springer.com/article/10.1007%2Fs10661-016-5310-7

Yihdego, Y. and Webb, J.A., 2017 "Assessment of wetland hydrological dynamics in a modified catchment basin: case of Lake Buninjon, Victoria, Australia". Water Environmental Research Journal Volume 89(2): 144-154. doi: 10.2175/106143016X14798353399331. https://doi.org/10.2175/106143016X14798353399331

Yihdego, Y, Reta, G and Becht, R., 2017 Human impact assessment through a transient numerical modelling on The UNESCO World Heritage Site, Lake Navaisha, Kenya. Environmental Earth Sciences 76: 9. DOI: 10.1007/s12665-016-6301-2. http://link.springer.com/article/10.1007/s12665-016-6301-2

Yihdego, Y 2016 Evaluation of Flow Reduction due to Hydraulic Barrier Engineering Structure: Case of Urban Area Flood, Contamination and Pollution Risk Assessment. Journal of Geotechnical and Geological Engineering, 34(5): 1643-1654. DOI:10.1007/s10706-016-0071-1. http://link.springer.com/article/10.1007/s10706-016-0071-1

Author Response

Response to Reviewer 1 Comments

Dear reviewer:

Thank you very much for your comments about our manuscript entitled “Evaluation of reservoir-induced hydrological alterations and ecological flow based on multi-indicators". We submit here the revised manuscript as well as a list of changes and the revised part has been marked yellow in the manuscript.

Mainly Changes:

  1. We have enriched “Introduction” part of the manuscript with the related articles that you provided.
  2. The discussion about the zero-flow situations has been added in section 3.5, and related suggestions have been put forward.

Thank you for the papers you have provided. We carefully read these papers which can give us new ideas and directions for our future work, such as the application of numerical models in ecological hydrology. Once again, thank you very much for your constructive comments and suggestions which would help us to improve the quality of the paper! If you have any question about this manuscript, please don’t hesitate to let us know. Hope these will make it more acceptable for publication.

Sincerely yours,

Dr. Mingqian Li

E-mail: hydrogeolmq@163.com

Reviewer 2 Report

This article has been corrected, several defects have been revised and research limitations have been revealed, and the overall quality is decent. Regarding the calculation of the Shannon index, citing studies conducted in East River, Yellow River, and Lake Shelbyville did not really answer the suitability of applying Eq5 to your study reaches. However, without the in-situ biodiversity data, maybe this is so far the best way to do it. One minor suggestion can be considered before publication.

1. Although there were no "zero-flow day" data before the SWR, the occurrence of zero-flow situations after SWR had a serious impact on river ecology. As the reach near the Liaoyang hydrological station often dries up for as much as 100 days owing to the large water demand, it would be appropriate to bring up this discussion in section 3.5.

Author Response

Response to Reviewer 2 Comments

Dear reviewer:

Thank you very much for your second comments about our manuscript entitled “Evaluation of reservoir-induced hydrological alterations and ecological flow based on multi-indicators". We submit here the revised manuscript as well as a list of changes (the replies are highlighted in blue and the revised part has been marked yellow in the manuscript).

Specific comments 1:

Although there were no "zero-flow day" data before the SWR, the occurrence of zero-flow situations after SWR had a serious impact on river ecology. As the reach near the Liaoyang hydrological station often dries up for as much as 100 days owing to the large water demand, it would be appropriate to bring up this discussion in section 3.5.

Response 1:

The discussion about the zero-flow situations has been added in section 3.5, and related suggestions have been put forward.

Once again, thank you very much for your constructive comments and suggestions, we really appreciate it! Hope these will make it more acceptable for publication.

Sincerely yours,

Dr. Mingqian Li

E-mail: hydrogeolmq@163.com

Round 2

Reviewer 1 Report

Pretty much my comments- addressed.
The content /flow of the manuscript has been improved (e.g. why the approach has been employed in the context of other alternative methods
and the implication of Zero- flow system with respect to the operation of reservoirs and river ecology impact).

This manuscript is a resubmission of an earlier submission. The following is a list of the peer review reports and author responses from that submission.

Round 1

Reviewer 1 Report

This article investigated the time series of eco-indicators, IHAs, and biodiversity to illustrate the impacts of dam construction. There is no serious flaw in this article. However, I was wondering if the calculation of SI (Eq.5) is suitable for the study river? Every river should have its ecological uniqueness. The authors should explain more reading the rationality to apply the SI, established somewhere else, to this study river. Besides, could the authors address more specifically how to avoid statistical redundancy by using eco-flow indicators (L36)? Moreover, could the authors provide suggestions regarding the operation of reservoirs to protect river ecology (L338)? The authors should address more about the status of the study watershed where SWR is not the only one infrastructure altering hydrology.

(Section 3.1.2) I would suggest to calculate the correlation coefficient between precipitation and eco-flow indicators separately for before and after the dam construction, demonstrating the effects of dam construction. What is the impact of the Guanyinge Reservoir, having a storage capacity of 21.68 x 108 m3, built in 1995 in the upstream of the Shenwo Reservoir? I can tell somehow something changed after 1995 in Fig.7 and Fig. 8, although I did not have statistical evidence. Did the growth of the water demand influence the indicators? There is the Tanghe Reservoir, which has a similar storage capacity as SWR and was built as early as 1969. in Section 2.1, I suggest not to use the term "natural conditions" and suggest to have more detailed description on the Tanghe Reservoir and the Guanyinge Reservoir. (Figure 8) Could the authors explain why the SI gradually increased in the beginning of dam construction in 1970 and reached SI maximum when the construction was complete in 1974? L249, L258. Should it be 32 IHAs? From other public reports and researches, the reach near the Liaoyang hydrological station often dry up for as much as 100 days owing to the large water demand, which is contradictory to what is addressed in the context.

Reviewer 2 Report

General comments

The datail of identifying the ecological impact is not clear. the ecological impact  uncertainity is way less than the errror introduced from other model paramtetrs.

The method emplyed in this paper (Index method) -section 2.3.3 might be limited to specific site condition. No-linearity among variables and paramters has not been well appreciated in the paper.

It would be advisable to consider other analytical and numerical modeling to compliment or justify the conclusions arrived. Please see suggested references below to consider alternative methods employed  (for hydrology regime alteration) elewhere.

Also the time to reach equlibrium (sytem) to attain steady state after impacts) needs more clarification. Moreover, direct, indirect and cumlative impacts are not explicitly mentioned either, perhaps due to the simplisitic method applied.